

# The influence of diurnal snowmelt and transpiration on hilllslope throughflow and stream response

Brett Woelber[1], Marco P. Maneta[1], Joel Harper[1], Kelsey G. Jencso[2], W.Payton Gardner[1], Andrew C. Wilcox[1], and Ignacio López-Moreno[3]

[1]Geosciences Dept, University of Montana, Missoula, MT
[2]W.A. Franke College of Forestry and Conservation, University of Montana, Missoula, MT
[3]Dpt. Procesos Geoambientales y Cambio Global. Pyrenean Institute of Ecology. C.S.I.C. Zaragoza, Spain

**Correspondence:** Marco Maneta (marco.maneta@umontana.edu)

**Abstract.** Daily stream flow and groundwater dynamics in forested subalpine catchments during spring are to a large extent controlled by hydrological processes that respond to the day-night energy cycle. Diurnal snowmelt and transpiration events combine to induce pressure variations in the soil water storage that are propagated to the stream. In headwater catchments these pressure variations can account for a significant amount of the total pressure in the system and control the magnitude,

duration, and timing of stream inflow pulses at daily scales, especially in low flow systems. Changes in the radiative balance at the top of the snowpack can alter the diurnal hydrologic dynamics of the hillslope-stream system with potential ecological and management consequences.

We present a detailed hourly dataset of atmospheric, hillslope, and streamflow measurements collected during one melt season from a semi-alpine headwater catchment in western Montana, US. We use this dataset to investigate the timing, pattern, and

linkages among snowmelt-dominated hydrologic processes and assess the role of the snowpack, transpiration, and hillslopes in mediating daily movements of water from the top of the snowpack to local stream systems. We found that the amount of snowpack cold content accumulated during the night, which must be overcome every morning before snowmelt resumes, delayed water recharge inputs by up to 3 hours early in the melt season. These delays were further exacerbated by multi-day storms (cold fronts), which resulted in significant depletions in the soil and stream storages. We also found that both diurnal snowmelt

and transpiration signals are present in the diurnal soil and stream storage fluctuations, although the individual contributions of these processes is difficult to discern. Our analysis showed that the hydrologic response of the snow-hillslope-stream system is highly sensitive to atmospheric drivers at hourly scales, and that variations in atmospheric energy inputs or other stresses are quickly transmitted and alter the intensity, duration and timing of snowmelt pulses and soil water extractions by vegetation, which ultimately drive variations in soil and stream water pressures.

*Copyright statement.* TEXT



# 1 Introduction

In snow-dominated dominated headwater catchments, hillslope hydrologic processes and streamflow during spring are largely determined by fluxes of snowmelt (Bales et al., 2006; Hood and Hayashi, 2015). The hourly magnitude of snowmelt varies regularly with the day-night cycle across a multi-month period (Flint et al., 2008). Snowmelt thus produces a regular sequence

of infiltration events into the soil that drive hillslope and stream hydrology, in contrast to the typically more irregular pattern of rainfall and associated hydrologic influence. A marked feature of hillslopes during the spring freshet, as opposed to rain-dominated systems, is the duration, extent and magnitude of hillslope saturation. As long as snow is actively melting, meltwater inputs significantly exceed evapotranspiration outputs, augmenting the soil water storage and generating an increasingly connected soil saturated layer across hillslopes (McNamara et al., 2005).

Diurnal streamflow fluctuations have been observed over a range of climates and scales (Czikowsky et al., 2004; Lundquist and Cayan, 2002) , especially with respect to water uptake by vegetation in riparian corridors (Butler et al., 2007; Gribovszki et al., 2008; Loheide II, 2008; White, 1932). Diurnal groundwater variations caused by snowmelt and/or in shallow groundwater systems (Gribovszki et al., 2010) are poorly understood by comparison. Diurnal fluctuations in the water table may have a limited impact on the overall water balance of a catchment, especially during high flow periods. However, increasing interest

in low-flow hydrologic processes and the relevance of these processes to water management, highlight the importance of more detailed study of diurnal cycles and other elements of links among snowpack, melt, and hillslope and stream hydrology. An analysis of the diurnal cycles provides insight into the processes that contribute to river discharge, especially in first order catchments, and help diagnose and anticipate streamflow increases or decreases for water management decisions (Lundquist and Cayan, 2002). For instance, diurnal fluctuations in the groundwater system are transmitted to streams and control the

magnitude, duration, and timing of stream inflow pulses, which can significantly affect fish habitats in low flow reaches. During snowmelt pulses, daily cycles inject cold, low-solute water into the groundwater system and the stream, altering the soil temperature, the entrapment of excess air, and the osmostic potential and solvent capacity of the soil water with consequences for its biogeochemistry (Lipson et al., 2002; Petrone et al., 2007; Zhao et al., 2017).

The study of diurnal cycles of snowmelt, water table, and stream recharge fluctuations in snow-fed river systems comple-

ments the study of diurnal streamflow signals in areas dominated by evapotranspiration Loheide II (2008); Wondzell et al. (2010) or glacial melt (Magnusson et al., 2014). Diurnal cycles of melt have been used to infer snowpack properties Caine (1992); Jordan (1983), and to obtain a hydrologic characterization of the soil and its connection to streams in high-elevation regionsKurylyk and Hayashi (2017); Loheide and Lundquist (2009a); Lowry et al. (2010); Magnusson et al. (2014). A diurnal evapotranspiration cycle may be superimposed to the snowmelt signal; both cycles combine at different intensities and phases

to produce a complex signal that propagates to the groundwater system and to streams. Solar radiation and air temperature are considered the two major cyclic forcings that induce thaw and evapotranspiration, exerting a strong control on soil moisture, recharge to the shallow groundwater system, and soil hydraulic gradients (Webb et al., 2015). Freeze and thawing conditions are common in high-elevation regions in spring, when daily temperature oscillate with an amplitude around $10^{\circ}$C, and maximum temperatures above freezing (Gribovszki et al., 2010). These are typical conditions observed in the spring in high elevation





region. Evapotranspiration process occurs during the growing season, defined as the period when minimum air temperatures stay above freezing (Kunkel et al., 2004). These conditions do not occur in high elevation regions until late spring. As the melt season advances, the snowmelt signal eventually fades, whereas the evapotranspiration cycle strengthens as days grow longer and incoming solar radiation increases. Few studies have investigated how both signals combine to produce diurnal hydrologic

responses in high-elevation catchments.

In this study, we present a comprehensive set of atmospheric, hillslope, and streamflow measurements from a semi-alpine headwater catchment in western Montana, US to investigate the timing, pattern, and linkages among snowmelt-dominated hydrologic processes. We recorded hourly atmospheric variables, sap flow velocity, water table elevation at five shallow wells and water stage in a perennial stream during one melt and growing season. To provide insight into the role of diurnal snowmelt

and transpiration on soil throughflow, hillslope storage, and stream response, we analyze pressure variations in the water table and the stream and thereby illustrate shifting flow pathways and their influence on streamflows. In addition we compare radiative exchanges at the surface of the snowpack and tree water uptake to daily groundwater and stream stage fluctuations in order to understand how diurnal cycles are generated and propagate to local stream systems.

## 2   Site description

Our study site is a small (2.9 km2) headwater catchment in the Bitterroot Mountains of western Montana at elevations spanning from 1950 to 2250 mASL. The site is in the Lost Horse Creek basin, which was formerly glaciated, has granitic bedrock (Foster et al., 2001), and flows west to east into the Bitterroot River (Figure 1). The site consists of a number of gentle (approximately 14°) north-facing hillslopes and steeper (20° to 25°) south-facing hillslopes covered by open areas of grass and shrub vegetation intermixed with stands of Douglas fir (Pseudotsuga menziesii), sub-Alpine fir (Abies lasiocarpa) and Engelmann spruce (Picea

engelmannii). Annual precipitation varies from 1270 to 1770 mm per year, the majority of which falls in the form of snow. Snow cover typically begins in late October, and persists to late June. Bedrock in this area is overlain by sandy to silty-loam soil as per the USDA textural classification system. At this site and in other areas in the sub-catchment, the depth to which soil can be excavated is roughly 50 cm. Geophysical analysis using ground penetrating radar reveal a 30 cm layer of saprolite under this soil layer and above un-weathered granite bedrock. Hydraulic conductivity is about 1.5 m/day in the soil and about 0.08

25  m/day for the weathered granitic bedrock (Woelber, 2013). The watershed is drained by a first-order perennial creek incised to the bedrock and with a  10 m wide riparian zone.

## 3   Methods

### 3.1   Instrumentation

We instrumented a 40 m long, north-facing hillslope at the bottom of the study drainage with a meteorological station, 5 shallow

monitoring wells (Figure 1), one stream stage probe, and 4 thermal dissipation probes to monitor sap flux in two trees. The meteorological station recorded precipitation (Campbell Scientific TE525), snow depth (Campbell Scientific SR50A), wind




velocity and direction (Campbell Scientific RM Young 05103-45), net radiation (Campbell Scientific NR-LITE), and incoming and reflected shortwave radiation (Campbell Scientific CS300). These sensors were positioned at a height of 4 m above the ground surface to remain above the snow surface. Atmospheric data from this station were recorded at 15-minute intervals. Snow water equivalent (SWE) was taken from the Twin Lakes snowpack telemetry (SNOTEL) station located 100 m from the

5 study site.

Five wells were placed in the anticipated path of groundwater flow along the study hillslope. These wells were placed with a drive-point rod and sleeve, pounded to the point of refusal, and installed with 12.7 mm diameter PVC encasing. Wells were backfilled with sand and capped with bentonite. These five wells were instrumented with HOBO pressure transducers (error +/- .3 cm), which continuously measured the pressure head of the saturated layer at 15-minute intervals. Stream stage

was monitored with another pressure transducer placed in a low-velocity pool that was continuously connected to the main stream draining our site. Measurements from both well and stream pressure transducers were calibrated with manual field measurements.

Sap flow velocity was monitored at 10-minute intervals through the stem of two adjacent Engelmann Spruce of different age using the Granier-type Dynamax Thermal Dissipation Probe (TDP-50) sap flow monitoring system (Dynamax, 1997; Granier,

1987). Three probes were installed in the larger Engelmann Spruce and one additional probe in an adjacent younger Engelmann Spruce. The trees were located about 20 m from the stream outside the riparian zone. Core samples of nearby tree trunks of same species and similar diameter were used to estimate the sapwood area and calculate sapflow volumes. However, in this paper only tree activity, as indicated by sap flow velocity, is reported.

### 3.2 Data preprocessing

The analysis focuses on one melt and growing season spanning from 1 April to 1 September 2012. All datasets except stream stage cover this period without gaps. The stream stage record starts on 26 April, once ice had melted sufficiently to permit pressure-transducer installation. All datasets were aggregated to hourly time steps by simple averaging to reduce noise and homogenized with a common hourly time stamp index. The sap flow velocity signal of the 4 probes were highly correlated (Pearson's r > 0.9); we therefore coalesced those data into a single pooled time series by simple averaging.

### 25 3.3 Conceptualization of the hillslope fluxes

Our analysis of the hillslope fluxes follows the work by White (1932) and later extended by Loheide II (2008) to estimate evapotranspiration from diurnal groundwater fluctuations. These authors observed that once the conditions that sustain transpiration cease, water flows laterally from the vicinity of the location where the water table fluctuation was observed to replace the water extracted by vegetation. Both authors considered that the incoming water came from a regional source far enough that it

was not affected by fluctuations. While White (1932) considered that the hydraulic head at the recovery source was constant, Loheide II (2008) improved this concept arguing that the rate of change of head at the recovery source was equal to the overall linear rate of water table change at the observed location. By subtracting this linear trend from the record of water table depths, the effect of incoming lateral flows is eliminated and the rate of decline reflects purely the rate of plant water uptake.



In our site the soil saturated layer is very local and inflows of water to recover pressure losses at a given location come from nearby, so unlike in Loheide's concept we expect that pressure variations at a point and at the recovery source covary. The effect of this is that the overall rate of change of the water table is highly variable and complex with periods of increase and decrease as the extent of the soil saturated layer grows and shrinks. Furthermore, in addition to diurnal fluctuations by plant

water uptake during the growing season, we also have to consider diurnal increases in the water table caused by snowmelt. In our case, then, the regional aquifer needs to act not only as a recovery source, but also as a sink of excess pressure.

Omitting deep percolation water losses to bedrock, the mass balance for the soil saturated layer at any point in the hillslope is

$$S_y \frac{dh}{dt} = f - ET + q_{in} - q_{out} = \Delta q_v + \Delta q_h \qquad (1)$$

where $S_y \equiv \frac{dS}{dh}$ is the specific yield, $h$ is pressure head, $t$ is time, $\Delta q_v \equiv f - ET$ are net vertical fluxes, $f$ is recharge from infiltration, $ET$ is evapotranspiration losses, $\Delta q_h \equiv \Delta q_{in} - \Delta q_{out}$ are net lateral (horizontal) fluxes, qin is lateral uphill incoming fluxes, and qout are lateral downhill outgoing fluxes. These last two terms in the right hand side of the mass balance represent the lateral water exchanges and control the general (multiday) trends of the water table on which diurnal fluctuations are superimposed. In the absence of $f$ and $ET$, when $q_{in} > q_{out}$ the elevation of the water table increases (flux convergence)

creating an upward trend in the groundwater record. Conversely, when $q_{out} > q_{in}$ the water table decreases (flux divergence), creating a declining trend.

We argue that the local time trend of the water table time series is an approximation of $\Delta q_h$. By subtracting it from 1, the detrended record reflects solely the net vertical fluxes $S_y \frac{dh_{DT}}{dt} = \Delta q_v$ , where $h_{DT}$ is detrended pressure head. To calculate the local trend of the water table and detrend the water level time series we used a centered, 24-hour moving average window. In

the context of analyzing daily fluctuation cycles, a 24-hour moving average window is most appropriate. Increasing the moving average window would be more likely to reflect the influence of lateral flows with longer travel distances in calculated trends. Detrending the time series using a moving window has been used in other studies (e.g., Graham et al., 2013), but only with the intention of enhancing and isolating the diurnal signal.

### 3.4 Data analysis

The hour of daily maxima was calculated from the detrended time series as the time the maximum detrended flow occurred each day. Because of the circular nature of hourly times series, a late diurnal peak of a given day can occur early in the morning of the following day. To solve this problem, and maintain ordinal consistency we unwrapped these peak times into the following day by adding 24 hours when peaks in the morning are preceded by peaks in the evening.

We also calculated rolling (time-evolving) correlation coefficients between daily total snowmelt and daily maximum ground-

water level, between maximum daily groundwater levels and maximum diurnal stream levels, between maximum daily transpiration and minimum daily groundwater level, and between maximum daily transpiration and minimum stramflow levels. We





used groundwater levels measured at well 4 because it records the clearest series of diurnal fluctuations. The rolling coefficients for each day were calculated with the points within a 16-day moving window centered at the specific day.

Finally, we plotted the empirical phase-space of the hillslope-stream system using the detrended time series of well 4 and streamflow. The components of the velocity vectors showing the trajectories in the phase-space plane were calculated by
differentiating the respective detrended pressure-head time series.

## 4  Results

### 4.1  Atmospheric drivers

Atmospheric forcing (Figure 2a,b) and the associated hydrologic response of the hillslope (Figure 2 c-i) exhibit clear diurnal and seasonal cycles. Maximum daily net radiation increased as spring advanced, peaking around the summer solstice, and then
declining (Figure 2a). Unlike maxima, minimum daily net radiation did not exhibit a clear seasonal cycle and was consistently negative, indicating that radiative cooling was happening at the surface during nighttime periods. During the study period (1 April – 1 September) precipitation inputs totaled 399 mm (Figure 2b), over half of it (224 mm) occurring during the three large cold fronts identified in the figure (56% of the total precipitation). Early in the season and during the cold fronts nighttime temperatures were consistently below freezing. These nighttime freezing events became infrequent after mid May.

### 4.2  Snowpack and transpiration

Figure 2c-i presents the hydrologic response of the hillslope. The elements that mediate the interaction between soil hydrology and atmospheric inputs are the snowpack, through the melt process, and the vegetation, through the transpiration process (Figure 2c). Other than two modest accumulation events early in the study period, the snowpack (snow water equivalent) monotonically decreased as energy inputs accumulated and snow melted (Figure 2c). Ablation rates were up to 3 cm/day early
in the melt season and up to 7 cm/day as days grew longer and warmer late in the melt season. During multi-day storms, daily snowmelt ceased, and snow water equivalent remained constant or modestly increased. The snow sensor location was snow-free in late June.

Transpiration— observed as relative sap velocity— increased from mid-May, when minimum daily temperatures were consistently above freezing and daytime periods grew sufficiently long (Figure 2c). Soil water uptake by vegetation is tied to
the radiation cycle and was highly sensitive to cloudy days. This is clearly seen during the third multi-day cold front, when transpiration was greatly reduced.

### 4.3  Hillslope response

The daily response of wells to melt inputs varied according to their relative position along the hillslope, but the seasonal response of all wells exhibits a similar pattern (Figure 2d-h). At the beginning of the period the snowpack was ripening and
outputs from the snowpack increased soil moisture, reducing the soil hydrologic deficit. However, a saturated layer in the



soil did not develop until the entire soil profile was at field capacity and could rapidly transmit gravitational water to the soil-bedrock interface. The first signal in the piezometers occurs during first multi-day storm (cold front 1). Variations in the soil-saturated layer during this early transient period were between 10 and 25 cm among all wells. Diurnal fluctuations in the piezometric response were highly dampened, indicating that the saturated layer was generated by rain inputs and snowmelt

induced by temperature or radiation inputs were of secondary importance. Groundwater levels at wells located uphill showed a greater drawdown, with well 1 (figure 2d) drawing down in excess of 25 cm. Water subsidies incoming from uphill positions dampened the recessions at wells located down the slope. The recessions are progressively less steep as the uphill contributing areas increase, with well 5 (Figure 2h) drawing down less than 10 cm. After this first cold front passed, snowpack ablation and water outputs into the soil did not resume immediately.

From May 10 to June 25, large sections of the hillslope were fully saturated. Because the soil water storage capacity was exhausted in some parts of the hillslope, diurnal snowmelt pulses did not contribute to the soil saturated layer. During this period melt pulses were less clearly recorded at some wells. Saturation conditions decreased during cold front 2 and cold front 3, which produced a refreezing of the snowpack, a cessation of snowmelt inputs, and a recession in the soil saturated layer. During these cold fronts water level diurnal fluctuations disappeared or were heavily dampened in all wells. From June 25 to

July 10, well levels receded when the snowpack melted out and water inputs into the hillslope ceased. As a result of upslope water subsidies, downslope wells decreased later and at a slower rate than upslope wells.

### 4.4 Stream response

During the melt season stream stage responded to diurnal energy inputs and to the level of soil saturation (figure 2i). Average stage increased and decreased with the amount of hillslope saturation. These variations disappeared as soon as the soil saturated

layer vanished, indicating that the most transient source of water to the stream was throughflow from the soil and that water in the weathered bedrock, which feeds the stream during the summer, is slow moving. Within the general trend of water levels in the stream, diurnal fluctuations are tightly coupled to those of the soil saturated layer. However, unlike soils, the storage capacity of the stream is not limited by soil depth, therefore stage fluctuations and diurnal cycles are very clear throughout the entire period. The exceptions are the three cold front periods. As the snowpack refroze and the melt pulses stopped, the stream

stage receded and diurnal fluctuations disappeared. During these periods inflows into the channel were purely lateral flows from the existing soil water storage. On the other hand, streamflow diurnal oscillations had more amplitude and were crisper when soils were at or near maximum saturation. Saturation-excess overland flow pulses generated from snowmelt during these periods were directly transferred to the channel with less dispersion.

When the water supply to the stream from the soil saturated layer ceased, soon after the snowpack melted, streamflow

quickly receded to its base flow levels (Figure 2cont). During this period, inflows into the stream were supplied by the low conductivity saprolite, maintaining a stream stage that declined very slowly (about half a centimeter in two months). Unlike the flows supplied by the soil saturated layer, flows from the saprolite do not have a discernible diurnal signal. Transpiration and air temperature cycles during the summer months maintain a strong diurnal cycle (Figure 2contb,c), but these are not apparent in the summer stream baseflows. This is an indication that unlike the soil, which was very sensitive to atmospheric diurnal cycles,



water in the bedrock was more uncoupled from atmospheric conditions. Note also that only a few summer storms produced a modest increase in stream flow stage. However, it is increasingly recognized the role that bedrock moisture has in sustaining plant transpiration (Rempe and Dietrich, 2018), so the absence of diurnal cycles in summer flows cannot easily be interpreted as an indication that evapotranspiration occurs mostly in the soil and to a lesser extent in the bedrock. Low bedrock diffusivity

is a damping mechanism that could make diurnal fluctuations unobservable in the stream

## 4.5    Timing of diurnal maxima

Detrended versions of the time series presented in Figure 2 represent the vertical component of throughflow in soils and streams, and the detrended data enhance high frequency fluctuations that are used to calculate the magnitude and timing of diurnal water table and stream stage oscillations (Figure 3). Early in the season daily piezometric and stream stage maxima

occurred late in the evening, between 2000 and 2400 local standard time (Figure 4). As the snowpack thinned and matured (Figure 4), water travel times within the snowpack decreased, shifting the diurnal peak response of the piezometers towards earlier in the day This shift of diurnal peaks toward earlier in the day was not apparent in the stream until June. By early to mid June diurnal peaks in some wells occurred as early as late afternoon. Around this time water uptake by vegetation increased (Figure 4). From June on snowmelt inputs continued to decline and the rate of water uptake by vegetation increased. At the

end of June and early July, when most snow melted out from the hillslope groundwater peaks on average occurred early in the morning but as the snowmelt signal faded and evapotranspiration and other water transmission factors gained influence, the timing was more variable as observed in the larger spread around the lowess curve in most wells (Figure 4a-e).

## 4.6    Interaction of transpiration and snowmelt diurnal pulses

The impact of daily transpiration losses on the soil saturated layer and on the stream, which would be expected to manifest

as gradual rises and sharp stage declines, are difficult to disentangle from the fluctuations caused by snowmelt pulses. Diurnal cycles caused by snowmelt have a sharp rise and a gradual decline (Jordan, 1983), while diurnal cycles caused by evapo-transpiration have a gradual rise and sharp decline Loheide and Lundquist (2009a); Lundquist and Cayan (2002). We did not observe such transition in the shape of the diurnal oscillations. Through the entire melt season the diurnal cycles recorded at the wells and the stream featured a sharp rise and a gradual decline, indicating that snowmelt pulses were dominant. The

transpiration signal was present although obfuscated by the superposition signals with changing phases and frequencies. The clearest indication that both diurnal cycles were present (snowmelt and transpiration) is the lenticular patterns generated by the oscillations in Figure 3. This structure is a standard beat interference resulting from the superposition of two periodic waves of slightly different amplitudes or when their phases are shifting at slightly different rates (Figure 5). This type of interference was observed in all piezometers and in the stream when the snowmelt and evapotranspiration signal gradually reinforced or

canceled each other out as the signals align or shift out of phase.

   Another clear indication that transpiration is extracting water from the soil was in its relationship with minimum daily flows. The magnitude of the daily snowmelt pulses were directly correlated with the magnitude of the water level maxima in the soil. Also, water level maxima were correlated with streamflow maxima. Similarly, maximum daily transpiration was inversely





correlated with diurnal minima in the soil and the stream (Figure 6). Water level maxima at well 4 correlated very well with both daily average snowmelt and water level maxima in the stream (Figure 6). These correlations, however, deteriorated gradually as the melt season advanced. Transpiration maxima also showed very strong negative correlations with water level minima at well 4 and the stream. These correlations quickly reverted to positive when the ground became snow free, and then were
reduced to zero when soil water contribution to the stream ended and base flows where sustained by water from the saprolite.

Early in the season, when the snowpack is deep, travel times were long and water levels in the ground and the stream peaked late in evening, out of phase with the afternoon of maximum transpiration activity, so transpiration and diurnal water level minima were highly correlated. As the season advanced and the snowpack thinned and densified, snowmelt travel times decreased and water levels peaked earlier, interfering with the afternoon minima of the transpiration signal and reducing the
correlations.

### 4.7 Hillslope storage and streamflow dynamics

Declining diurnal snowmelt pulses were associated with a decline in the amplitude of the diurnal groundwater pulses. In general, groundwater oscillations had larger amplitude during the first half of May, but a trend can only be evaluated in Well 4 because the soil at the other wells were close to saturation during significant parts of June and did not fully register
diurnal snowmelt events during much of the study period. Because amplitudes are truncated in these wells, we did not analyze their oscillations. At well 4 (Figure 8a) the recorded oscillations confirm that as the melt season advanced and the snowpack thinned, the amplitude of the diurnal signal in the saturated soil layer decreased. By mid June interferences caused by the intensification of the transpiration signal, with a different phase than snowmelt pulses, may have further reduced the amplitude of the combined signal. On the other hand, in the stream the amplitude of the oscillations was largest from early to mid June,
which was the period when the soil saturated area was at its largest extent, as discussed earlier, and diurnal snowmelt pulses had more opportunity to travel faster and with less dispersion as saturation-excess overland flow. The amplitude of diurnal stream fluctuations declined rapidly form mid June on as the snowpack melted out and the extent of the saturated area decreased.

Assuming that fluxes are proportional to water levels in a roughly linear way, the ratio of the amplitude of the diurnal signal to the local trend can be used to approximate the contribution of vertical fluxes to the total mass balance in the soil or stream
(Figure 8, blue line). In early May a fully connected saturated layer quickly developed and received water from uphill regions, as indicated by the slower recession of groundwater levels during cold front 1 (Figure 2g,h). Therefore, because the amount of throughflow in the lower part of the hillslope is larger, the vertical water fluxes accounted for a relatively small share of the total fluxes. In general, as May advanced and a connected saturated layer consolidated along the entire hillslope, vertical fluxes accounted from 5% to 25% of the total groundwater flux at Well 4. In June the soil was fully saturated in many sections
of the hillslope and oscillations caused by snowmelt infiltration and transpiration pulses were not registered in most wells or the net contribution of these vertical fluxes was negligible. During this period we expected relatively higher contributions of snowmelt inputs to the stream in the form of overland flow, and that contributions from overland flow would increase the amplitude of diurnal oscillations in stream water levels. This expectation is supported by the data presented in Figure 8b. In the stream diurnal water fluctuations were consistently between 3% and 7.5% of the total water levels (Figure 8b), with the lowest



contributions during cold front 2 and cold front 3 when snowmelt stopped. The largest contributions of diurnal fluctuations occurred early in the season, when water depths in the stream were relatively low, and during June when relatively large amounts of snowmelt were quickly transferred to the stream as overland flow. Overland flow transfers the pulses less abated by diffusion than subsurface flow, so the amplitude of the oscillations increased and peaked during June, when the extent of

saturation in the hillslope was largest (Figure 8b, red dots). The amplitude receded when the extent of saturation in the soil started to wane in July.

## 5   Discussion

### 5.1   Diurnal pulses and water levels

Radiative exchanges are a major driver of diurnal cycles in high-elevation environments, controlling snowpack energetics and

transpiration. Stream recharge from diurnal melt events had a prompt, same-day response to radiative forcing indicating a strong atmosphere-snowpack-soil-stream hydraulic connection very sensitive to the day-night energy exchanges at the top of the snowpack. Turbulent energy exchanges (i.e. latent and sensible heat) between the snowpack and the lower atmosphere in semi-alpine environments are arguably of secondary importance to radiative exchanges (Sicart et al., 2004, 2006), especially early in the melt season when temperatures are still low. During the study period, wind speeds at the below-canopies location of

the sensor rarely exceeded 2 m/s and typically values were lower than that, providing a weak forcing for turbulent exchanges. Under the assumption of radiative dominance, negative net radiation early in the season indicates that the snowpack increased its cold content and refroze at nighttime (Figure 2b). This energy deficit needed to be recovered daily before snowmelt resumed. Long nights and late season storms increase the cold content of the snowpack, often add a new layer of snow that decreases the albedo of the surface, and add dry porosity that needs to be saturated before water output occurs. These processes impose a

snowpack 'energy hurdle' that needs to be overcome, which delays the resumption of the water output process (Seligman et al., 2014). As the melt season advanced and the nights grew shorter, the time taken to overcome the energy hurdle produced by nighttime radiative losses and to resume the daily snowmelt cycle shrank on average by 0.75 min/day (Figure 9). The total time needed to overcome the energy hurdle (energy replenishing time) ranged from 200 minutes in mid-April to 100 minutes in early July. This calculation aligns with the delays of water outflows measured under snow pillows (Webb et al., 2017). Given the

relatively short days in April, a delay of more than 3 hours takes a significant proportion of the effective daily melting period. During each of the three multi-day storms observed during the 2012 melt season, the role of daily energy replenishing time in delaying melt is clearly illustrated. During these colder time periods, negative radiative outputs during the night were not exceeded by positive radiative inputs during the day and snowmelt did not resume. During these periods snow water equivalent levels did not change, well levels declined, and stream stage decreased.

An earlier onset of snowmelt output caused by a declining nighttime energy deficit does not explain why water table and streamflow daily maxima peaked earlier as the melt season advanced. Rather, changes in snowpack thickness and residual saturation, changes in the thickness and saturation of the soil unsaturated zone, and the continuity of the soil saturated layer determine how quickly water moves from the top of the snowpack to the stream. In the sub-alpine hillslope instrumented for





this study, the unsaturated zone was minimal or non-existent during the spring melt when hillslopes were mostly or entirely saturated. Therefore, the reduction in the delay from the moment the snowpack was isothermal and saturated to the moment when groundwater peaks occur from May to early June (Figure 3a-e) was caused by the gradual snowpack thinning and the gradual increase in the hydraulic conductivity of snow as the snowpack densifies and saturates (Caine, 1992; Jordan,

1983). The daily water table maxima from early May to June roughly went from late in the evening to mid afternoon with a lot of variability (Figure 3a-e). Some wells respond on a daily basis while others only respond some days, suggesting that some landscape positions receive and store daily inputs of meltwater while others remain at saturation over multiple days and immediately transmit meltwater in the form of overland flow.

Despite the variability in the timing among the 5 wells, they show a timing pattern very similar to that in Jordan (1983), who

found that peaks in a semi-alpine watershed in British Columbia occurred at around midnight early in the melt season, when the snowpack is thickest, shifting to earlier in the day as the melt season progresses, and arriving as early as 1500 when the snowpack was about 15 cm deep. Gribovszki et al. (2006), in contrast, found that the elevation of the water table during the melt season in a low-elevation experimental catchment in Hungary correlated with air temperature, with afternoon maxima and early morning/dawn minima. This discrepancy highlights differences between high- and low-elevation snowmelt-influenced

catchments. In high-elevation catchments with deep snowpacks, where radiation is a main driver of the energy balance, longer travel times within the snowpack and the time required to make the snowpack isothermal and ripe result in substantial delays with respect to peak energy inputs. At lower elevations where snowpack may be shallower and warmer, temperature is more likely to dominate the daily melt cycle, resulting in faster water transfers to the groundwater system better synchronized with diurnal temperature oscillations.

Once the daily water pulse is in the saturated layer of the soil, it moves downhill toward the channel. Although the five piezometers were installed following the downhill gradient and at relatively short distances from each other, we did not find clear patterns or consistent lags in the timing of peaks between the five wells in the transect. This may reflect water losses by seepage into the bedrock and the multiplicity and evolving dynamics of groundwater flow paths in the soil-bedrock interface (Freer et al., 1997; Maneta et al., 2008; Freer et al., 2002). Unlike groundwater levels, the timing of diurnal streamflow peaks

during May started at around 2200 local time and continued peaking late in the evening until June (Figure 4f). The increasing hydrological connectivity within the hillslope and the associated increase in flows into the stream was the most probably what maintained water levels peaking late in the evening. It was not until June that stream water levels started to peak earlier. Timings of diurnal streamflow peaks reported in the literature are variable. The lags depend on the source of the fluctuations and the way they propagate, although the exact mechanisms are still unclear (Graham et al., 2013). In general, peaks are reported

from late in the afternoon (Mutzner et al., 2015) to late in the evening or early in the morning depending on the scale of the watershed, and on delays and travel times from the positions when melt occurred Loheide and Lundquist (2009b); Lundquist et al. (2005); Lundquist and Cayan (2002). The timing of diurnal peaks is typically about noon early in the season and tend to shift progressively toward later in the day as the relative saturation of the watershed increases, and the snowline and active melt band of the snowpack shifts toward higher altitudes changing and lengthening the flow paths (Lundquist and Cayan, 2002).

However, shifts to earlier times have been reported for small rivers when travel times within the snowpack dominate (Jordan,





1983) and when transmission velocities through the soil and the channel system increase as hydrologic connectivity in the hillslope-channel system develops and water is routed faster (Wondzell et al., 2007).

Regardless of the source, diurnal fluctuations only appeared in the stream when a saturated layer was present in the soil. Correlations between transpiration and snowmelt fluxes and water level extrema in the soil and the stream, as well as recognizable

interference patterns in the diurnal signals provided evidence of the superposition of both signals, but we did not find more direct indications of the balance of individual contributions such as recognizable changes in the symmetry or the emergence of multimodality in water level diurnal cycles. No diurnal stream fluctuations were observed in summer flows, when the saprolite supplied water to the stream, suggesting that either the main source of water for transpiration was the soil and uptake from the saprolite was minimal, or that the hydraulic diffusivity of the saprolite was sufficiently small to dampen the amplitude

of the oscillations below the precision of the transducer in the stream. Our results indicate that disentangling the individual contributions of diurnal input and uptake fluxes in the response of a semi-alpine hillslope is difficult due to the complexity of the interaction between diurnal and seasonal forcing factors driving transpiration and snowmelt cycles.

## 5.2  Hillslope-Stream dynamics

Diurnal fluxes accounted for between 2% and 12% of the pressure variations in the stream, the remaining being contributions

from older water stored in the soil and saprolite. These diurnal contributions are on the lower end of the 5% to 25% reported for high-elevation rivers in the western US during the melt season (Lundquist and Cayan, 2002), although large variations should be expected given the site specific nature of the factors that reapportion the mass and energy balance of a hillslope at local scales. A convenient way to visualize and discuss the behavior of the hillslope-streamflow system is by plotting the empirical approximation of its phase space (Figure 9). This type of diagram presents the dynamics of the states (hydraulic

head) at which the hillslope and the stream are at each moment in time, and the trajectories the system follows in the recorded sequence of states. Empirical approximations from data are noisy and do not have the typical groomed aspect of the phase spaces generated by differential equations, but still provide substantial qualitative insight into the system dynamics. In Figure 9 detrended hydraulic heads at Well 4 and at the stream represents the hillslope-stream system. The system has one stable fixed point at (0,0), which represents daily average pressure head at Well 4 and at the stream. The system relaxes to this

point when external forcings cease. The trajectories of the system oscillate about this point in counterclockwise damped orbits caused by the periodicity of the snowmelt and transpiration boundary conditions. The damping is mainly caused by changes in the boundary conditions but also by changes in the soil absorptivity as discussed below. Given by order of axes -Well 4 pressure head represented in the x axis and stream pressure head in the y axis- the counterclockwise sense indicates that diurnal groundwater oscillations peaks earlier and drives streamflow peaks. Positive relative pressures in the hillslope and the

stream are typically found in the late afternoon and evenings (upper right quadrant of graph), while negative relative pressure both in the hillslope and the stream tend to be found in the late morning and early afternoon. Early in the season the amplitude of diurnal oscillation at Well 4 were larger than oscillations in the stream, orienting the orbits toward the x-axis. As the season advanced and the oscillations at Well 4 decreased and became less ample than the oscillation in the stream, the orbits became aligned with the y-axis.



The amplitude of the oscillations in the hillslope was dampened as the season progressed, because the recovery of cold content in the snowpack during nighttime decreases and reduces the period when melt is shutoff, but also by changes in the absorptivity and specific yield of the soil unsaturated layer as evapotranspiration dries the soil. The hydraulic capacitance (specific moisture capacity) of wet soil is relatively low compared to dry soils, which magnifies the oscillations of soil water levels in response to diurnal inputs/outputs. As the unsaturated zone dries and its specific yields and soil absorptivity increase, oscillations in the soil saturated layer are damped and the trajectories in Figure 9 become dominated by oscillation in the stream. This damping mechanism in which water is transferred from the soil saturated layer to the soil unsaturated layer was analyzed by Duffy (1996) and later by Brandes et al. (1998) using low-dimensional numerical models, and showed that tradeoffs between the soil saturated and unsaturated storages introduce nonlinearities in the rainfall-storage-discharge relationships at the hillslope scale, especially when the unsaturated zone is dry. However, when the unsaturated zone is wet, the damping is reduced. This is significant because under reduced damping diffusive effects become secondary to gravity in driving water fluxes. In fact, Brandes et al. (1998) show that lower frequency oscillations in soil storage (multiday lengths) not induced by an external periodical forcing are possible under wet conditions and heavy rains. Although we did not find evidence of these type longer wavelength oscillations in our measurements, it is clear that transient oscillations in the soil saturated layer can be easily induced, especially under wet conditions.

The periodic water level trajectories presented in Figure 9 never completely relax to equilibrium, forming nearly closed orbits driven by the solar cycle. This behavior can only be attributed to the existence of an external forcing cycle and the nonlinearity of the recharge terms (snowmelt and transpiration). Without such forcing only the unlikely existence of a positive damping mechanism that pumps in the energy lost to friction and viscous forces each cycle, could produce closed orbits. However, transient oscillatory behavior can persist well after the forcing ends, especially if the damping mechanisms are weak, producing tight spiral trajectories resembling closed orbits (Brandes et al., 1998; Duffy, 1996).

## 6 Conclusions and implications

Snow-dominated headwater catchments are vulnerable to alterations in energy and precipitation regimes, highlighting the importance of understanding linkages between local atmospheric, hillslope, and fluvial processes. This is especially true for smaller low-flow regime systems. In this paper we presented and analyzed a unique, high-temporal resolution observational dataset that contributes to advancing our understanding of how diurnal snowmelt and transpiration cycles drive the hydrology of a snow-dominated, semi-alpine headwater catchment during the melt and growing seasons.

The snowpack mediates the response of the monitored system to atmospheric drivers by accumulating cold content at night and during cold multiday storms, and becoming a energy sink during the day until such deficit is replenished. Overcoming this energy hurdle delays the production of snowmelt into the soil by several hours, especially early in the melt season. The freeze and thaw cycles create pulse-like infiltration events that induce pressure oscillations in the soil storage system, which are subsequently transmitted to the stream. Accordingly, as the thickness of the snowpack declines with the melt season, the travel times of melt water to the base of the snowpack and to the soil saturated layer are reduced, shifting diurnal pressure





peaks progressively to early in the day. Changes in the energy balance at the surface of the snowpack can potentially reduce melt response times, and accelerate the hydrologic response of headwater catchment, with potentially ecological and hydrologic implications not only for first order catchments, but also for the timing of spring freshets in higher order downstream hydrologic systems.

Even in a small, well connected system, the interactions between diurnal solar cycles, snowmelt and the response of water levels in the hillslope and the stream showed high variability and complexity. With the onset of the growing season, the timing and dynamics of soil and stream water oscillations is further altered by diurnal plant water uptake cycles. The interactions of the diurnal snowmelt cycle and the transpiration cycle in the production of diurnal pressure fluctuations in the soil and stream system are difficult to disentangle because snowmelt and transpiration cycles are correlated and driven by the same solar cycle.

We did not find differences in the shape of snowmelt-driven and transpiration-driven soil and stream water oscillations, as it was reported in the literature for studies conducted in riparian zones and lower elevation catchments. However, we interpreted the beat-like patterns in the observed oscillations around mean daily soil and stream pressures as evidence of the interaction between pressure oscillations induced by these two periodic signals (snowmelt and transpiration). The existence of this inter-action is further corroborated by a correlation analysis between snowmelt and transpiration cycles and diurnal maxima and

minima in soil and stream storages.

Snowmelt and transpiration pulses are tied to the rotation of the earth, and while the period of these cycles are not expected to change significantly in the future, their intensity can change with alteration in the amount of energy inputs from the atmosphere. Our analysis showed that the hydrologic response of the snow-hillslope-stream system is highly sensitive to atmospheric drivers at hourly scales, and that variations in atmospheric energy inputs or other stresses are quickly transmitted and alter the intensity,

duration and timing of snowmelt pulses and soil water extractions by vegetation, which ultimately drive variations in soil and stream water pressures. From our analysis it was also clear that at small scales (well to hillslope), our high-elevation semi-alpine hydrologic system did not damp high frequency pressure variation as quickly as the local diffusive nature of flow in porous media may have lead us to believe. Furthermore, we did observe the soil damping mechanisms varied in intensity during the season, and we discussed that when the unsaturated one is wetter, pressure oscillations are propagated with less

attenuation. Clearly, the speed and damping of pressure variations in system will depend on the soil physical characteristics and the geometry of the hillslope, but we show here that high-frequency pressure oscillation in the hillslope and stream systems can be easily induced and that these may play an important environmental role that is overlooked.

Although these pressure variations are small in absolute terms, they are a significant portion of the total hydraulic head in soils and streams of low flow systems and control, among other things, the arrival of water pulses, which maintain the

perennial nature of some first order streams and extend the upslope extent of riverine systems (Godsey and Kirchner, 2014). Alterations in these water pulses may decrease the connectivity of habitats, disrupts the exchanges of energy and materials in high-elevation riverine ecosystems, and may destroy or restrict access to refugia used by fish and other species. Furthermore, these pressure variations control the patterns of arrival of colder, low-solute snowmelt water into the soil system and the stream, which may change the soil water temperature and osmotic potentials with consequences for the biochemistry of the soil, and

the quality of water for soil and riverine biota. Improving our understanding of high-frequency variation in the hydrology of





high elevation catchments may also help us identify shifts in hydrologic behavior that can be used as tell tales of upcoming changes that can propagate to higher order systems. These are important questions of relevance for management that can only be answered through a better understanding of how the hydrology of this type of environments work and respond to high frequency (subdaily) inputs. This study complements the study of diurnal hydrologic fluctuation in shallow groundwater and

5   stream systems, most of which focus on low elevation semiarid regions, riparian zones, or only address hydrologic fluctuations induced by transpiration or by snowmelt.

*Code and data availability.*   The dataset and a Jupyter notebook with the analysis performed for this stud can be downloaded from https://bitbucket.org/maneta/Brettetal2018

*Competing interests.*   The authors declare that they have no conflict of interest.

10   *Acknowledgements.*   This material is based upon work supported in part by the National Science Foundation EPSCoR Cooperative Agreement IIA-1443108 and EPS-1101342.



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





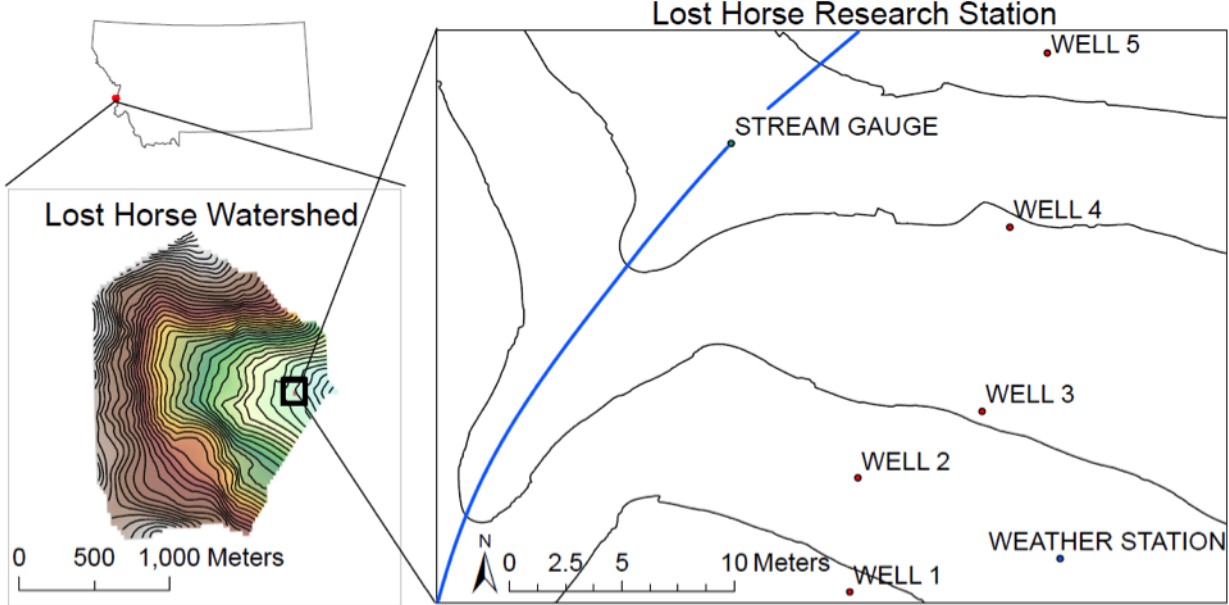

**Figure 1.** Study area in upper Lost Horse Creek watershed, including location in western Montana (upper left), topography (lower left), and instrumentation setup.

Zhao, Q., Chang, D., Wang, K., and Huang, J.: Patterns of nitrogen export from a seasonal freezing agricultural watershed during the thawing period, Science of The Total Environment, 599-600, 442–450, https://doi.org/10.1016/J.SCITOTENV.2017.04.174, http://www.sciencedirect.com/science/article/pii/S0048969717310185, 2017.





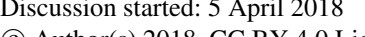

**Figure 2.** Time series of atmospheric inputs and hydrologic states; a) net radiation, $\mathrm{Wm^{-2}}$; b) precipitation, $\mathrm{mmd^{-1}}$; c) snow water equivalent (blue line), cm, and relative sap velocity (red line), $\mathrm{cmh^{-1}}$; d-h) pressure head in piezometers 1 to 5 (blue line), cm; i) pressure head in stream (blue line), cm. The grey shaded area indicates three major cold storms during the study period.



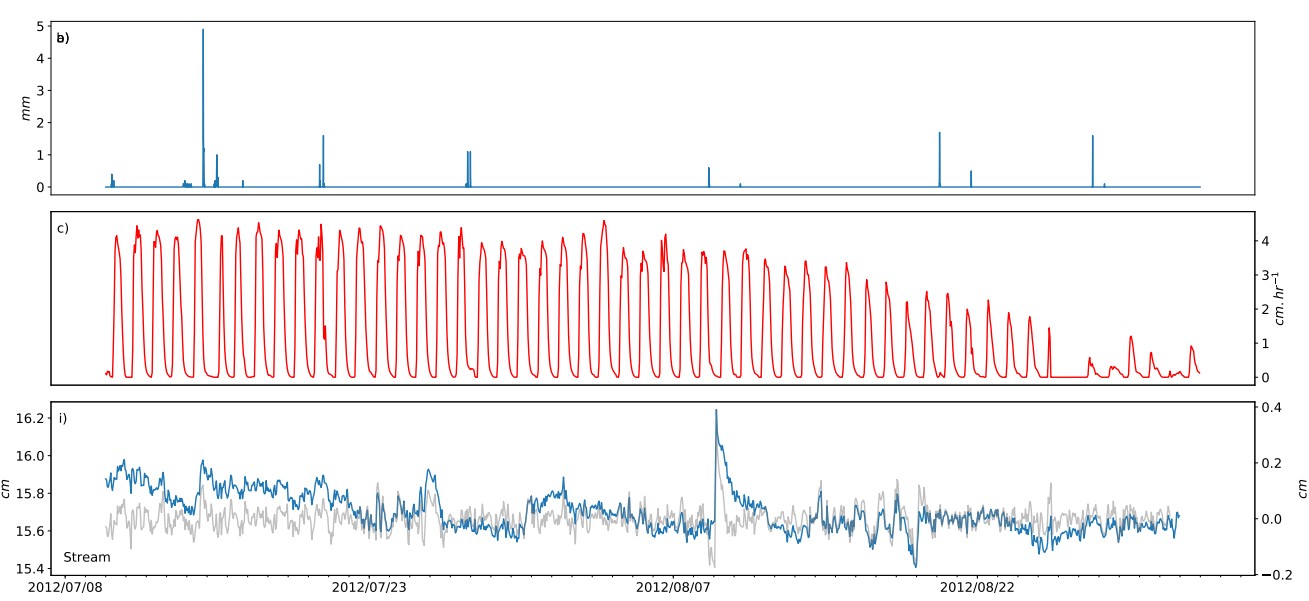

**Figure 2cont.** Same as figure 2 but for precipitation, relative sap velocity and pressure head in the stream during the low streamflow summer period. Letters are consistent with Figure2: b) precipitation, $\mathrm{mmd^{-1}}$; c) relative sap velocity, $\mathrm{cmh^{-1}}$; and i) pressure head in stream (blue line), $\mathrm{cm}$, and detrended pressure head in stream (grey line).







**Figure 3.** Detrended water levels for wells 1 to 5 (a-e) and for the stream (f) during the melt season. The grey shaded area indicates three major cold storms during the study period.





**Figure 4.** Black circles indicate the hour of the day when daily pressure peaks occur in wells (a-e) and the stream (f). Ordinate time of day unwraps the hours into the following day when peaks in the evening get followed by peaks in the morning (e.g. 1am gets unwrapped to be hour 25 if previous days had evening peaks). The black dashed line is the lowess fit of the timing of peaks. For context, the figure background contains the SWE (blue line) and sap relative velocity (red line) as shown in Figure 2.



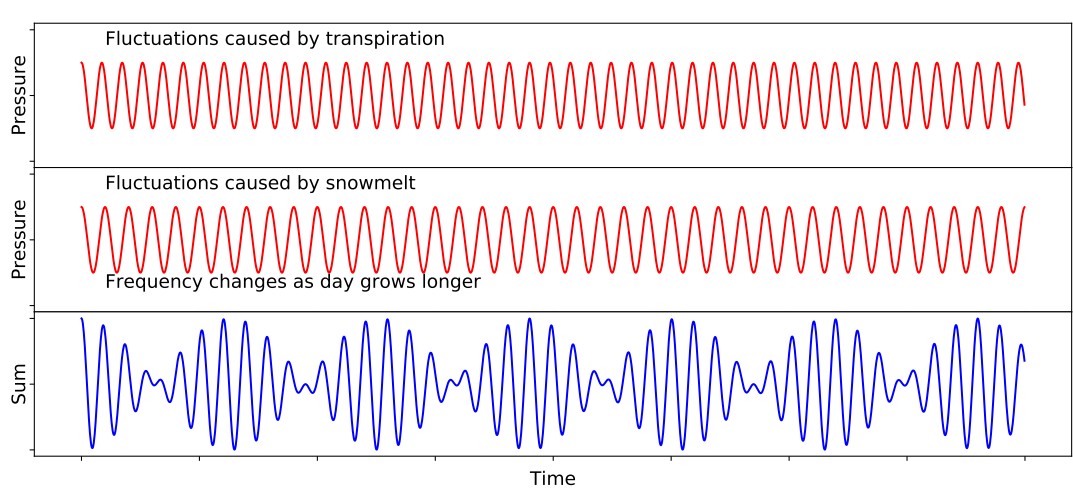

**Figure 5.** Additive interference of two sinusoidal pressure signals, one caused by transpiration (top) and other by snowmelt (middle), with slightly different frequencies. As the season advances and the days grow longer, the frequencies of transpiration and snowmelt will change. The resulting signal (bottom) is the sum of both waves. The amplitude of the envelope of the resulting signal is sinusoidal with a frequency determined by half the difference of the frequency of the individual signals. Compare with Figure 3.





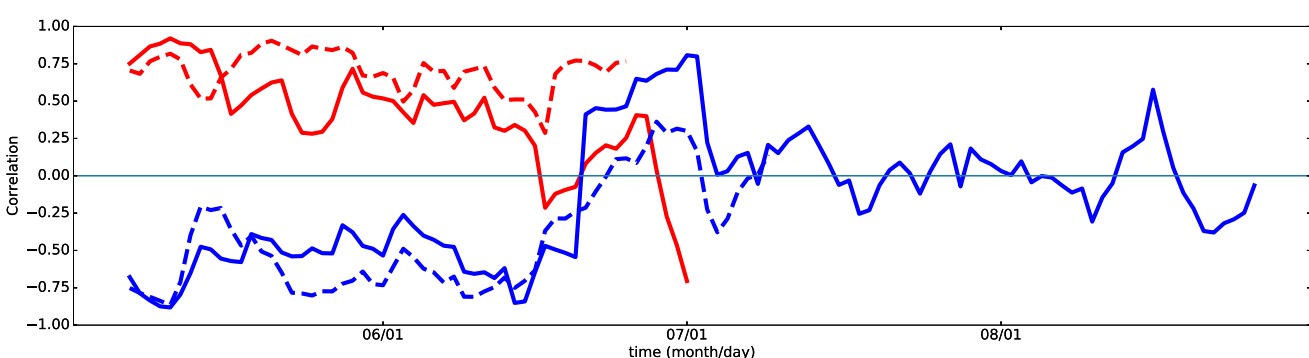

**Figure 6.** Moving correlations between the following variables: Red dashed line: total diurnal snowmelt and maximum daily water level at Well 4; Red solid line: maximum daily water level at Well 4 and maximum water level in stream; Blue dashed line: maximum daily transpiration and minimum daily water levels at well 4; Blue solid line: maximum daily transpiration and minimum water level at stream. Well 4 was chosen because it contains the largest number of diurnals. Correlations for each day were performed using the points within a centered 16 day window.





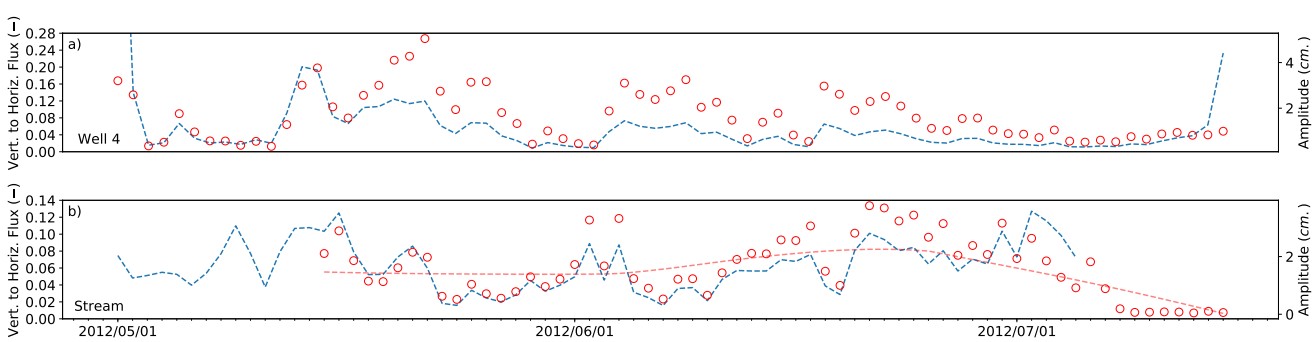

**Figure 7.** Blue dashed line, left axis: Estimation of the contribution of diurnal (vertical) fluxes to lateral (horizontal) fluxes at Well 4 (a) and stream (b); Red circles, right axis: Amplitude of diurnal pressure head fluctuations for Wells 4 (a) and stream (b). The dashed red line in Figure 8b is the lowess fit of diurnal amplitudes in the well.



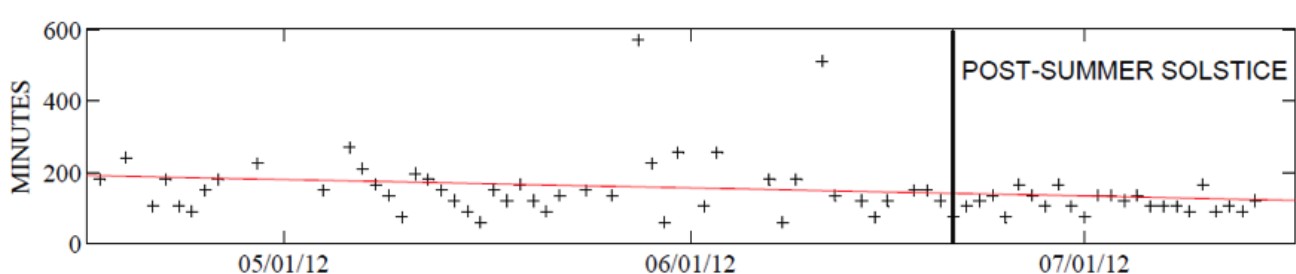

**Figure 8.** Daily replenishing time during the melt season. Each marker indicates the time taken by daily radiative inputs to compensate nightly radiative losses during the previous night. This calculation provides an estimate of the amount of time required each day to return the snowpack to the water output phase. The summer solstice is highlighted with a vertical line.





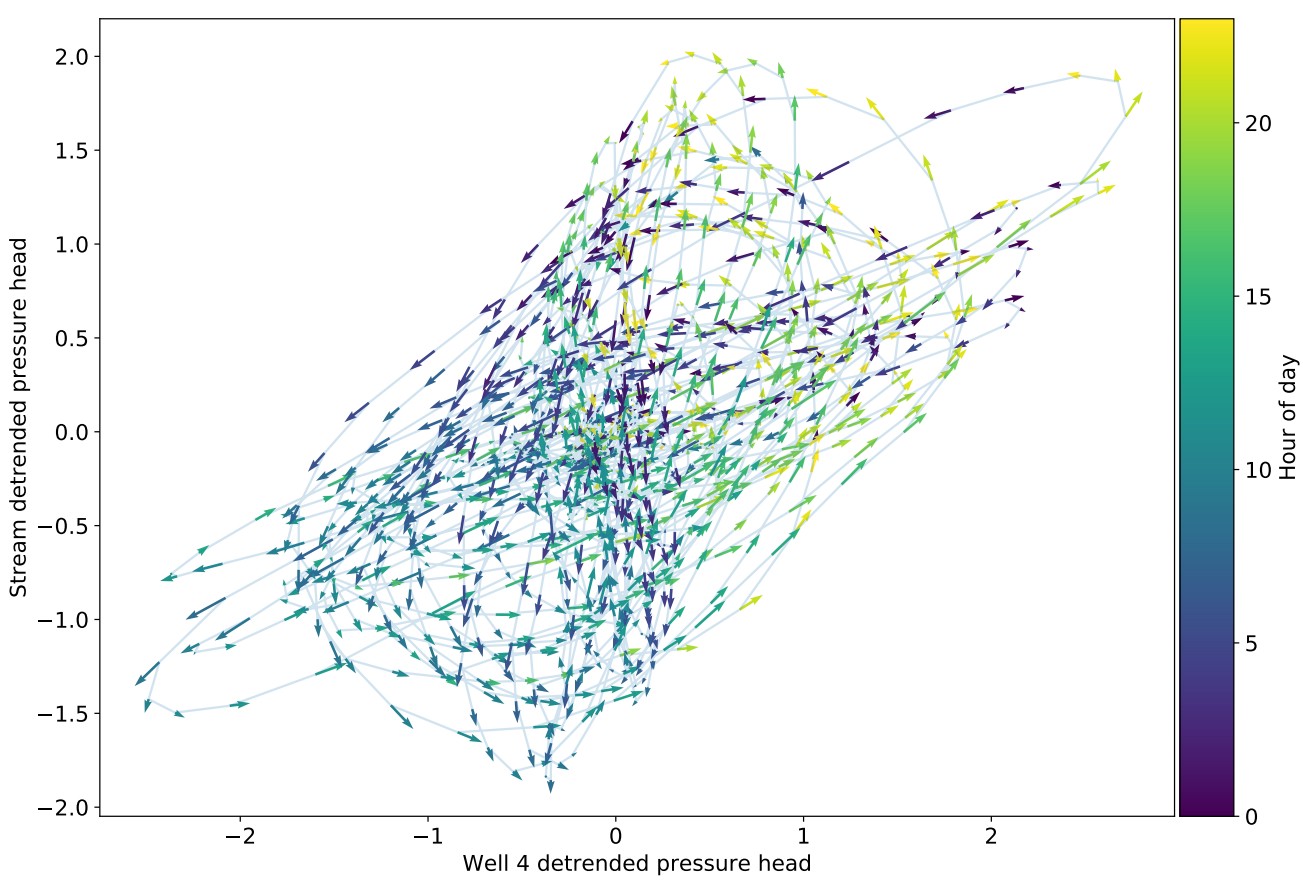

**Figure 9.** Reconstruction of the phase portrait of the hillslope-stream system from the record of observations. The states are the detrended hydraulic heads at Well 4 and at the stream. The lines and the vectors indicate the trajectory and velocity of the states. Colors indicate the time of day at which the system was at that corresponding state.