# Peer review of "The influence of diurnal snowmelt and transpiration on hillslope throughflow and stream response"

_Hydrology and Earth System Sciences, 2018_

## Referee Comment (RC1) · Anonymous Referee #1 · 12 May 2018

Woelber et al. have collected a very nice dataset of diurnal fluctuations in wells, streams and sapflow, and they have a descriptive presentation of their measurements and discussion of related publications in the field. However, I fail to see how this paper advances our scientific understanding of the processes involved. I would recommend that the authors formulate clear hypotheses and focus their work to specifically test these hypotheses. For example, if they want to investigate the snowpack energy balance related to the diurnal timing, they should run an energy balance snow model and explicitly test this process and then use their data to support or falsify their hypotheses. There is potential in this dataset, but further work is needed before publishing a paper.

---

## Referee Comment (RC2) · Anonymous Referee #2 · 23 May 2018

[referee-annotated manuscript omitted]

---

## Author Comment (AC1) · 24 Jun 2018

Comments from Reviewer #1:

Woelber et al. have collected a very nice dataset of diurnal fluctuations in wells, streams and sapflow, and they have a descriptive presentation of their measurements and discussion of related publications in the field. However, I fail to see how this paper advances our scientific understanding of the processes involved. I would recommend that the authors formulate clear hypotheses and focus their work to specifically test these hypotheses. For example, if they want to investigate the snowpack energy balance related to the diurnal timing, they should run an energy balance snow model and

explicitly test this process and then use their data to support or falsify their hypotheses. There is potential in this dataset, but further work is needed before publishing a paper.

Response from Woelber et al.:

We thank Reviewer 1 for their time reading the paper. Our work provides significant observational insight into how diurnal energy cycles drive small-scale pressure variations in a snow-dominated hillslope-stream system and contributes to our understanding of how the snowpack mediates hydrologic behavior at the hillslope and subdaily time-scales. Although streamflow variations induced by diurnal snowmelt and transpiration pulses has been studied before, how these processes evolve and combine to generate the observed diurnal hillslope and stream response is still unclear and warrant more research (Graham et al., 2013). We are presenting an observational study, along the lines of many other published studies (e.g. Lundquist and Cayan, 2002, Caine, 1992, Hood and Hayasi, 2015), not a modeling paper. However, to guide the analysis and answer the research questions we used a simple conceptual modeling framework that extends that proposed by Loheide (2008), which has proved very useful to understand small-scale diel hydrologic processes. Although we disagree with the notion that complex models are necessary to test hypotheses and advance science, we agree that running an energy-balance snow model, as suggested by the reviewer, has merit and could yield new insights. That approach is well beyond the scope of our study, however, and would result in a different paper from the one we submitted for review.

To address the suggestion that we more clearly articulate the research questions and hypotheses, as well as the specific insights contributed by the paper, we have revised the paper, especially the introduction and conclusions. The modifications are highlighted in the revised paper attached as supplemental review materials. However, for completeness we reproduce here the specific research questions we address in the paper:

How does the snowpack mediates the interaction between atmospheric inputs and

hillslope response in snow-dominated regions? Does night-time snowpack refreezing significantly alter the timing of water pulses into streams? How much and in what form do direct snowmelt and vegetation water uptake pulses contribute to producing the observed diurnal pressure variations in the soil saturated layer? and, How are water pressure variations induced by the diurnal solar cycle transmitted from the hillslope to the stream?

We start our research from the assumption that in our study site, during the melt season, diurnal variations in the snowpack energy state, as well as alterations in flow timings and pathways induced by changes in snowpack depth and density, are larger controls on the amount and timing of diurnal and seasonal water inputs into streams than transpiration or contributions from the mountain block aquifer. This is highlighted in the last paragraph of the Introduction in the revised manuscript. This initial assumption is the implicit working hypothesis that guided the research. The evaluation such assumption led us to quantify the relative contribution of snowmelt pulses to total diurnal pressure variations in the hillslope-stream system and to interpret seasonal variations in the timing of diurnal pressure peaks. The manuscript is also driven by a methodological hypothesis, which is now explicitly stated in the first paragraph of section 3.3 of the revised manuscript (see supplemental review materials). This conceptual model assumes that pressure dynamics in the soil saturated layer are the sum of separable horizontal (hillslope throughflow) and vertical (snowmelt, transpiration) water fluxes. Furthermore, we posit that local daily pressure variations are induced by superimposed cyclic snowmelt and evapotranspiration pulses. Interference in the signals should be observed in the resulting pressure variations as the amplitude and phase of input signals are modified by processes that delay and alter the duration of snowmelt and evapotranspiration, such as nighttime snowpack refreezing, snowpack depth, or changes in soil absorptivity.

Direct answers to the research questions indicated above are that in our study site, soil-water pressure fluctuations are dominated by the snowmelt signal and that the effect

of transpiration is more subtle. We did not observe the change in the shape of diurnal pressure waves reported in other studies when groundwater fluctuations transition from being snowmelt controlled to being evapotranspiration controlled. However, an apparent beat interference pattern in the soil-water pressure signal emerges as indication of the interaction of both input signals. Another relevant results is that cold content accumulated in the snowpack by nighttime radiative losses can delay the production of snowmelt up to 3 hours, especially early in the melt season, which is an important portion of the daytime period. Furthermore, we use our modeling framework to quantify the relative contribution of vertical fluxes to total hillslope throughflow in our study area, which can be up to 20%, and the role of soil absorptivity in damping the propagation of these pressure pulses toward the stream. These and other results directly answering the research questions are more clearly listed in the Conclusions. To increase clarity we have moved the ecological and management implications of these results out of Conclusions.

Finally, we want to point out that the manuscript contains methodological details and field methods that can be useful to guide the experimental design and data analysis of similar studies. We hope that these changes aiming at increasing the specificity of the paper goals address the reviewer concerns about the value of the paper and the specific research questions and hypotheses we investigated.

[Figure]

**Supplement:**

[revised manuscript text omitted]

---

## Author Comment (AC2) · 24 Jun 2018

Comments from Reviewer #2

This is a very good manuscript that presents and analyzes a dataset of snowmelt, groundwater levels and stream stage measurements. The authors present analyses of stage a level variations, focusing on amplitude, phase shifts and so on. In general, there a few shortcomings in the manuscript. The material is presented coherently, the figures are of very good quality, and the discussion is supported by a clearly presented conceptual model.
The more frequent comments I have pertain to the organization of the text, as sometimes the authors mix result presentation with discussion. Also, grammar should be checked, as there are a few instances of mixed-up singular/plural noun and verb usage. Please see the attached annotated manuscript for specific comments.

Overall, I recommend accepting with minor reviews.

Please also note the supplement to this comment: https://www.hydrol-earth-syst-sci-discuss.net/hess-2018-166/hess-2018-166-RC2-supplement.pdf

Response to Reviewer #2

We thank reviewer #2 for their time in reviewing the manuscript. We appreciate their positive assessment of the work and the general and specific comments that improve the organization of the manuscript and its readability. To facilitate the review process and to clearly identify the actions taken to address the reviewer comments, we reproduce below, line by line and in bold letters, the specific annotations the reviewer left on the pdf version of the original manuscript, along with the corrections/changes made (in italics).

Page 2, line 28: Please verify citations format. All citations have been verified to comply with the journal style.

Page 2, line 34: A word is missing here. Fixed.

Page 4, line 21: Did the stream cross section remain stable (unchanged) during the season? In many alpine sites, sediment transport in spring can be significant and alter the discharge/stage relationship? We did not use a stage-discharge curve for this study, we only analyzed the variations in streamflow stage. Nevertheless, the channel cross section remained stable during the study period, during which peak flows were moderate. The creek is incised to the stable underlying bedrock and during our visits to download the dataloggers we did not notice changes in the banks or any significant accumulation of sediment. In the revised manuscript (page 4, line 18 of supplemental

review materials) we indicate that the channel cross-section around the measurement point remained stable during the study period.

Page 5, line 6: Is it appropriate to talk about a "regional" aquifer in this geographical setting? We mean the aquifer in the consolidated rock, which is of a larger extent than the perched saturated layer in the soil-bedrock interface. In the revised manuscript we have changed "regional aquifer" by "mountain-block aquifer".

Page 5, line 9: Later in the manuscript you introduce a discussion on hydraulic properties of unsaturated/saturated soils and how they change in time. I think this discussion should be introduced much earlier, probably here. We believe you are referring to the discussion on the possible emerging beat interference in the diurnal pressure signals. Following this suggestion, we have moved the conceptual figure and the associated explanatory discussion to this section (see changes to section 3.3. in supplemental review materials).

Page 5, line 11: Something is not quite right in the definition of terms: when working out the variables in eq. 1, it comes to $dq\_h = q\_in - q\_out$. Thus, the inline equation in line 11 is mathematically wrong. The inline equation defining qhqin-qouthad incorrect s in the qin qout and has been corrected. Also, storage (S) is in dimensions of [L] (volume per unit area), such that SydSdhis dimensionless specific yield or drainable porosity. In the revised manuscript we specify units to make this clear (page 5, lines 23-26 of supplemental review materials).

Page 7, line 15: This is true except for well 4, which shows a more sensitive behavior than well 3. It'd be great to discuss why well 4 might be more sensitive overall. Soil at well 4 is significantly deeper than at any of the other wells and is less susceptible to saturation. This additional storage permits this well to record diurnal snowmelt events when other wells are saturated and register constant pressure head. Although the final recession in late June seem to start a little earlier than at well 3, situated immediately uphill, the recession follows the expected downhill progression from wells

1 to 5 in which draining rates are lower and saturation last a little longer as upslope accumulated length and water subsidies increase. However, well 4 also seems to lose pressure faster during periods of no snowmelt indicating higher draining capacity. The reasons for this are hard to determine because the exact subsurface water flow paths are unknown. The most likely explanation is that at the location of well 4, bedrock permeability is higher and the soil loses water to the bedrock aquifer at a faster rate. A second possibility is that horizontal conductivity is higher and downhill drainage is faster. A sentence discussing this is added in the revised manuscript (page 8, lines 1-4 of supplemental review materials).

Page 8, lines 2-5: This belongs in the discussion section. This sentence has been moved to the last paragraph of discussion section 5.1.

Page 8, lines 19-20: This sentence should come after the cited references... or maybe delete altogether as it is redundant with what comes afterwards. The sentence was redundant and is eliminated in the revised manuscript.

Page 8, lines 27-28: I believe this figure and the entire conceptual explanation should be presented before showing your actual measurements. We have revised the manuscript to introduce this figure and a conceptual explanation in section 3.3, which describes the conceptualization of the hillslope model.

Page 9, lines 6-10: This should go in discussion This paragraph was unnecessary and is removed in the revised version of the manuscript.

Page 9, line 23: In the soil? In the stream? Both in the soil and in the stream. We clarify this in page 10, lines 10-11 of supplemental review materials. This approximation is justified by the relatively small variations in stage induced by diurnal fluctuations.

Page 11, line 31: Check citation formatting. All citations have been verified to comply with the journal style.

Page 12, lines 10-12: Earlier you provided a convincing conceptual approach of superimposed signals. In your results, you measure these signals individually (snowmelt, stream, and et)... please state more clearly why it is difficult to disentangle this interaction, even though seemingly all intervening variables are measured directly. Even though we observe clear diel cycles of radiation and transpiration, and we observe clear cycles in the hillslope storage, we could only infer indirectly the joint effect of both signals. We weren't able to directly observe or attribute the contributions of the individual input signals (snowmelt and ETP) to pressure fluctuations. It is clear, however, that the snowmelt cycle dominates the resulting signals. As soon as the soil water drains out, the diurnal cycles in the stream disappear, which complicates attribution. A further complication is that the strength of both signals are very different. While a good snowmelt day can input into the soil amounts of water in the order of tens of mm (30-90 mm per day), transpiration takes out of the soil water amounts in the order of a few millimeters (2-4 millimeters per day). In the supplemental review materials (page 12, lines 28-34) we clarify why we say that disentanglement and attribution is complex:

"[...] we did not find more direct indications of the balance of individual contributions such as recognizable changes in the symmetry or the emergence of multimodality in water level diurnal cycles. A reason for this is that the strength of snowmelt and transpiration signals are different. Water inputs into the soil from snowmelt in a typical day are of the order of tens of mm (30-90 mm per day), however transpiration uptakes out of the soil water amounts in the order of a few millimeters (2-4 mm per day). With an extensive snow pack on the ground, snowmelt fluxes dominate diurnal hillslope storage fluctuations. This, and the varying interaction between the signals due to the shift in their timings, make it difficult to directly observe or determine the individual contributions of diurnal water inputs and uptakes on the observed hillslope response."

page 12, line 29: Check grammar. Grammar corrected, thanks.

Please also note the supplement to this comment:
https://www.hydrol-earth-syst-sci-discuss.net/hess-2018-166/hess-2018-166-AC2-

supplement.pdf

**Supplement:**

[revised manuscript text omitted]